# Effective Dereverberation with a Lower Complexity at Presence of the Noise

**Fengqi Tan, Changchun Bao * and Jing Zhou**

Speech and Audio Signal Processing Laboratory, Faculty of Information Technology, Beijing University of Technology, Beijing 100124, China
* Correspondence: baochch@bjut.edu.cn

**Abstract:** Adaptive beamforming and deconvolution techniques have shown effectiveness for reducing noise and reverberation. The minimum variance distortionless response (MVDR) beamformer is the most widely used for adaptive beamforming, whereas multichannel linear prediction (MCLP) is an excellent approach for the deconvolution. How to solve the problem where the noise and reverberation occur together is a challenging task. In this paper, the MVDR beamformer and MCLP are effectively combined for noise reduction and dereverberation. Especially, the MCLP coefficients are estimated by the Kalman filter and the MVDR filter based on the complex Gaussian mixture model (CGMM) is used to enhance the speech corrupted by the reverberation with the noise and to estimate the power spectral density (PSD) of the target speech required by the Kalman filter, respectively. The final enhanced speech is obtained by the Kalman filter. Furthermore, a complexity reduction method with respect to the Kalman filter is also proposed based on the Kronecker product. Compared to two advanced algorithms, the integrated sidelobe cancellation and linear prediction (ISCLP) method and the weighted prediction error (WPE) method, which are very effective for removing reverberation, the proposed algorithm shows better performance and lower complexity.

**Keywords:** speech enhancement; microphone array; dereverberation; noise reduction; linear prediction

## 1. Introduction

In real applications, reverberation and noise often cause the performance of speech communication or human–computer interaction to be degraded. Especially, due to the presence of reverberation, the speech signal [1] is often distorted, which seriously affects the localization of the sound sources. With the rapid development of artificial intelligence, human–computer interaction becomes increasingly more important [2,3]. If the reverberation and noise are not well cancelled in front of the intelligent devices, then speech communication will be greatly impaired.

In recent years, microphone array-based methods using spatial information have received extensive attention. The microphone array-based methods, such as the MVDR-based beamformer [4], the linear-constrained minimum variance (LCMV) beamformer [5,6], and the generalized sidelobe canceller (GSC) [7], are often used for speech enhancement. The microphone array-based beamforming methods can be used to receive signals from the direction of the target speaker and suppress the interference, such as reverberation and noise [8–10]. In [8], the GSC and a post-filter implemented with the Wiener filter were introduced to eliminate the reverberation and noise. In [9], the LCMV was used for the dereverberation in the case of multiple speech sources, in which the obtained sparse matrix was used to obtain a preliminary estimation of the desired signal and the LCMV was used to obtain the final estimation of the desired signal. In [10], a beamforming estimator based on minimum mean square error (MMSE) was proposed to obtain the early components of the reverberation signal. In MVDR beamforming, a relatively early transfer

function (RETF) was used. The beamforming and post-filter were used to manage the late reverberation and noise. The methods that only use beamforming to reduce reverberation have limited performance and cannot suppress the reverberation very well.

Currently, the most effective dereverberation method is based on multichannel linear prediction (MCLP) deconvolution [11–16], which is also known as the direct inverse filtering algorithm. The MCLP-based methods model the reverberation components as a delay of the speech signal in the time domain, namely the linear prediction components. In these methods, the estimation of the room impulse response (RIR) is not required, and the reverberation is subtracted from the speech signal itself. Thus, this method can be applied to the situation where the RIR cannot be reliably estimated. In [13], a statistical model-based dereverberation method was proposed. It assumes that the speech source follows the time-varying Gaussian model. The reverberation is modeled as the delayed components after linear prediction, where linear prediction coefficients are approximated by estimating model parameters to obtain the late reverberation components. This modeling method is defined as the weighted prediction error (WPE). In [11], by combining WPE and MVDR, the noise and reverberation were well suppressed simultaneously. The methods of using Kalman filter to estimate linear prediction coefficients for obtaining reverberation components were also proposed [12,14,15]. In [14], the MVDR and the MCLP were used for simultaneous noise reduction and dereverberation, and the Kalman filter was used to estimate the beamforming coefficients and linear prediction coefficients simultaneously. This kind of dereverberation framework that combines beamforming and the MCLP has attracted increasingly more attention. In [15], two interacting Kalman filters were used to alternately estimate the MCLP coefficients and reverberation signal. In [16], a two-stage algorithm was proposed. In the first stage, the WPE was used to reduce reverberation, and in the second stage, the MVDR was utilized to remove the residual reverberation. This method treats the reverberation as a diffuse noise. The Kalman filter-based dereverberation algorithm that combines the MCLP and the GSC, known as the integrated sidelobe cancellation and linear prediction (ISCLP), has also shown good performance [12]. This method not only reduces the complexity of simply cascading the MCLP and beamforming, but also improves the performance. However, in the ISCLP, estimating linear prediction coefficients with the Kalman filter needs to consider the initialization of the Kalman filter parameters [17–19]. These parameters, such as the power spectral density (PSD) of the target signal, are critical to the performance of the algorithm. Some related dereverberation methods [20–23] also heavily rely on the PSD estimation of the speech signal.

This paper proposes a dereverberation algorithm by combining the complex Gaussian mixture model (CGMM)-based MVDR (CGMM-MVDR) beamforming [24] and the linear prediction Kalman filter. In this algorithm, the spectral model based on the CGMM is used to estimate the time–frequency masks of the covariance matrix of the target speech and the steering vector. The covariance matrices of the target speech and noise are estimated through the time–frequency masks while considering the noise only, where the eigenvector corresponding to the maximum eigenvalue of the covariance matrix of the target speech is an estimation of the steering vector of the MVDR beamforming. At the same time, the main diagonal elements of the covariance matrix of the target speech are considered as the estimated PSD of the target speech, which are also required by the Kalman filter. Since this method can obtain a more reasonable estimation for the parameters, the performance of dereverberation is better than the ISCLP. In addition, the low-complexity version of the proposed method is proposed as well, in which the multichannel linear prediction coefficients are expressed in terms of the spatial filter and the interframe filter by using the Kronecker product. By determining the spatial filter, the Kalman filter is used to estimate the interframe filter. Since the dimension of the interframe filter is lower than that of the original multichannel linear prediction filter, the size of matrices in the Kalman filter also decreases so that the amount of calculation is greatly reduced. Experiments show that this method can effectively reduce computational complexity and obtain relatively good results.

In the following parts, we arrange the paper as follows. In the Section 2, we briefly introduce the signal model of the CGMM-based MVDR beamformer and the related linear prediction model. In the Section 3, we present the proposed method by combining the CGMM-based MVDR beamformer and Kalman filter. In the Section 4, we show and analyze the experimental results. Finally, we conclude our method.

## 2. Signal Model

In this paper, one sound source is considered, and $M$ microphones are used to collect the observation signal containing the reverberant and noisy speech signal. By performing a short-time Fourier transform (STFT), the received signal of the $m$th microphone can be expressed as $y_m(k,l)$ in the frequency domain, where $k$ and $l$ are the indices of the frequency bins and frames, respectively. The entire stacked microphone signals vector is denoted as follows:

$$\mathbf{y}(k,l) = [y_1(k,l), \cdots, y_m(k,l), \cdots, y_M(k,l)]^T \tag{1}$$

where superscript "$T$" indicates the transpose of the matrix. Thus, the multimicrophone signals corrupted by reverberation and noise can be written as follows:

$$\mathbf{y}(k,l) = \mathbf{x}(k,l) + \mathbf{v}(k,l) \tag{2}$$

where the vectors $\mathbf{x}(k,l)$ and $\mathbf{v}(k,l)$ are the STFT results of the speech signal containing reverberation and the noise signal, respectively. Here, the speech and noise are assumed to be mutually independent. Since the algorithm is finished independently in each frequency bin, the index $k$ is omitted for the simple expression in the subsequent parts.

### 2.1. Complex Gaussian Mixture Model

Reviewing MVDR beamforming based on the CGMM [24], the enhanced speech signal by the beamforming can be expressed as follows:

$$x_{\mathrm{b}}(l) = \mathbf{w}_{\mathrm{b}}^H \mathbf{y}(l) \tag{3}$$

where superscript "$H$" denotes the conjugate transpose, $\mathbf{w}_{\mathrm{b}}$ is the vector of the filter coefficients for minimizing the output power of the beamforming under the constraint $\mathbf{w}_{\mathrm{b}}^H \mathbf{g} = 1$. Here, $\mathbf{g}$ is the steering vector of the desired speech signal and the subscript "$_{\mathrm{b}}$" indicates the beamforming. Considering the sparsity of the speech signal in the time–frequency domain [25,26], the observed signal can be classified into three cases, i.e., the mixture of speech, noise, and reverberation; the mixture of speech and reverberation; and noise-only. Thus, the observation signals of the microphones can be written as follows [24]:

$$\mathbf{y}(l) = \mathbf{g}^{(d)} x_{\mathrm{b}}^{(d)}(l) \tag{4}$$

where the superscript "$d$" denotes the scenarios of the interference such as reverberation with noise (in this case, $d$ is denoted as x + r + n), reverberation-only (in this case, $d$ is denoted as x + r), and noise-only (in this case, $d$ is denoted as n).

The complex Gaussian distribution of the random variable $x$ with the mean $\mu$ and variance $\sigma^2$ is generally expressed as follows [24]:

$$\mathrm{N}(x; \mu, \sigma^2) = \frac{1}{\pi \sigma^2} \mathrm{e}^{-\frac{|x-\mu|^2}{\sigma^2}} \tag{5}$$

Assuming that $x_{\mathrm{b}}^{(d)}(l)$ follows a complex Gaussian distribution $x_{\mathrm{b}}^{(d)}(l) \sim \mathrm{N}(0, \phi(l)^{(d)})$, namely, its mean and variance are zero and $\phi(l)^{(d)}$, respectively. Here, $\phi(l)^{(d)}$ also indicates

the power spectral density of $x_{\text{b}}^{(d)}(l)$. Thus, at each time–frequency bin, the multichannel microphone signals obey the following complex Gaussian mixture distribution [24]:

$$\mathbf{y}(l)\,|\mathrm{d} \sim \mathrm{N}\left(0, \phi(l)^{(\mathrm{d})}\mathbf{R}^{(\mathrm{d})}\right) \tag{6}$$

where $\mathbf{R}^{(d)} = \mathbf{g}^{(d)}\,(\mathbf{g}^{(d)})^H$. Thus, the multichannel microphone signals can be described by the CGMM in the case of reverberation with noise or noise-only, in which the maximum likelihood estimation of the CGMM can be obtained by the expectation maximization (EM) algorithm [24].

The Q function in the EM algorithm is defined as follows [24]:

$$Q(\Theta) = \sum_{l}\sum_{d}\lambda(l)^{(d)}\log \mathrm{N}\left(\mathbf{y}(l); 0, \phi(l)^{(d)}\mathbf{R}^{(d)}\right) \tag{7}$$

where $\lambda(l)^{(d)}$ denotes the mask or probability of the different scenarios, which can be obtained as follows:

$$\lambda(l)^{(d)} = \mathrm{N}\left(\mathbf{y}(l); 0, \phi(l)^{(d)}\mathbf{R}^{(d)}\right) \Big/ \sum_{d}\mathrm{N}\left(\mathbf{y}(l); 0, \phi(l)^{(d)}\mathbf{R}^{(d)}\right) \tag{8}$$

In order to achieve the online MVDR beamforming, the speech signal is divided into a sequence of batches. Each batch contains a limited number of the speech frames. The number of the frames can be changed. Let $c \in \{1, 2, \dots, C\}$ denote the batch index and let $l_c$ denote the time index within the $c$th batch, where $C$ is the number of the batches. For each batch, the parameters of the CGMM are updated by the following formulation [24]:

$$\phi(l)^{(d)} = \frac{1}{M}\mathrm{tr}\left\{\mathbf{y}(l)\mathbf{y}(l)^H\left[\mathbf{R}_c^{(d)}\right]^{-1}\right\} \tag{9}$$

$$\mathbf{R}_c^{(d)} = \frac{A_{c-1}^{(d)}}{A_{c-1}^{(d)} + \sum\limits_{l \in l_c}\lambda(l)^{(d)}}\mathbf{R}_{c-1}^{(d)} + \frac{1}{A_{c-1}^{(d)} + \sum\limits_{l \in l_c}\lambda(l)^{(d)}}\sum\limits_{l \in l_c}\lambda(l)^{(d)}\frac{1}{\phi(l)^{(d)}}\mathbf{y}(l)\mathbf{y}(l)^H \tag{10}$$

where $A_{c-1}^{(d)}$ denote the sum of the masks over all the previous frames, and it can be updated as follows:

$$A_c^{(d)} = A_{c-1}^{(d)} + \sum_{l \in l_c}\lambda(l)^{(d)} \tag{11}$$

It is worth noting that $\mathbf{R}_c^{(d)}$ is usually irreversible. In practical calculation, the inverse of $\mathbf{R}_c^{(d)}$ is usually replaced by a pseudo inverse.

For every batch, the EM algorithm can be used to obtain the most optimal mask value for the current time–frequency bin only containing noise, and the covariance matrix in the case of the reverberation with noise or noise-only at each frequency bin can be estimated as follows [24]:

$$\mathbf{M}_c^{(d)} = \frac{A_{c-1}^{(d)}}{A_{c-1}^{(d)} + \sum\limits_{l \in l_c}\lambda(l)^{(d)}}\mathbf{M}_{c-1}^{(d)} + \frac{1}{A_{c-1}^{(d)} + \sum\limits_{l \in l_c}\lambda(l)^{(d)}}\sum\limits_{l \in l_c}\lambda(l)^{(d)}\mathbf{y}(l)\mathbf{y}(l)^H \tag{12}$$

So, the covariance matrix in the case of reverberation can be given as follows:

$$\mathbf{M}_c^{(\mathrm{x+r})} = \mathbf{M}_c^{(\mathrm{x+r+n})} - \mathbf{M}_c^{(\mathrm{n})} \tag{13}$$

Performing the following eigenvalue decomposition

$$\mathbf{M}_c^{(\mathrm{x+r})} = \mathbf{P}\mathbf{\Lambda}\mathbf{P}^{-1} \tag{14}$$

where **P** is the matrix composed of the eigenvectors of the matrix $\mathbf{M}_c^{(x+r)}$, and $\mathbf{\Lambda}$ is the diagonal matrix containing the eigenvalues, the eigenvector associated with the largest eigenvalue is extracted as the steering vector required for the MVDR beamforming. The coefficient vector $\mathbf{w}_b$ of the MVDR filter can be expressed as follows [24]:

$$\mathbf{w}_b = \left[\mathbf{M}_c^{(n)}\right]^{-1}\mathbf{g} \Big/ \left\{\mathbf{g}^H\left[\mathbf{M}_c^{(n)}\right]^{-1}\mathbf{g}\right\} \tag{15}$$

In fact, the CGMM-MVDR is a beamforming-based method with blind estimation of the steering vector. The late reverberation leads to an inaccurate estimation of the steering vector. Since the blind estimation of the accurate steering vector from the reverberated speech with noise is a very difficult task, what we obtain here is the approximation of the steering vector. This does not mean that such approximations lead to false results, because the reverberation has been mostly removed by Kalman filtering rather than beamforming.

### 2.2. Multichannel Linear Prediction Reverberation Model

By modeling reverberation components using the MCLP and predicting reverberation components of the current frame using the signals coming from the previous $(L - D)$ frames, where $D$ and $L$ are the frame numbers prior to the current frame, respectively, the speech signal with reverberation can be expressed as follows [12]:

$$\mathbf{x}(l) = \underbrace{\sum_{n=D}^{L} \mathbf{B}_{r,n}(l)\mathbf{x}(l-n)}_{\mathbf{r}(l)} + \mathbf{x}_e(l) \tag{16}$$

where $\mathbf{B}_{r,n}(l)$ denotes the multichannel linear prediction coefficients, and $L > D \geq 1$. The reverberation is divided into two parts: the early reverberation and the late reverberation. The result $\mathbf{r}(l)$ represents the late reverberation components. Here, the desired signal $\mathbf{x}_e(l)$, which is the direct signal with early reverberation components, is also called the prediction error in the linear prediction model. The prediction error is expressed as $\mathbf{x}_e(l) = [x_e^1(l), x_e^2(l), \cdots, x_e^M(l)]^T$, which contains the desired signal of each microphone. Note that $\tau = (L - D)$ is the number of the delayed frames that are used to predict the late reverberation components. The selection of $\tau$ should make the correlation between the late reverberation and the expected signal approach zero.

### 3. Proposed Dereverberation Method

The block diagram of the proposed dereverberation method is shown in Figure 1. In the upper path, the CGMM-based MVDR beamformer is used to enhance the input signal $\mathbf{y}(l)$ collected by the microphones so that we can obtain the enhanced signal $x_b(l)$. The PSD of the desired signal used for the Kalman filter is approximately equal to the power spectral density $\phi_{x_b}(l)$ of the $x_b(l)$. In the lower path, according to the delayed signals $\mathbf{t}(l)$ of the input signal $\mathbf{y}(l)$, the MCLP is performed based on the Kalman filter to estimate the reverberation signal $z(l)$ by linear prediction. The input to the Kalman filter-based linear predictor is $\mathbf{t}(l)$, and the corresponding output is the estimated reverberation signal $z(l)$. By subtracting $z(l)$ from $x_b(l)$, we can finally obtain the enhanced signal $s(l)$. Different from ISCLP [12], this paper uses the MVDR beamforming to replace the GSC, because the GSC has the shortage of speech distortion, that is, if the steering vector of the desired speech used in the GSC is not accurate, the desired speech will be inevitably destroyed by its block matrix. In particular, in the ISCLP, the PSD of the target signal is estimated by a complex algorithm, which contains some prior information [12], whereas the proposed method uses a simpler approach without any prior information to estimate the PSD of the target signal.

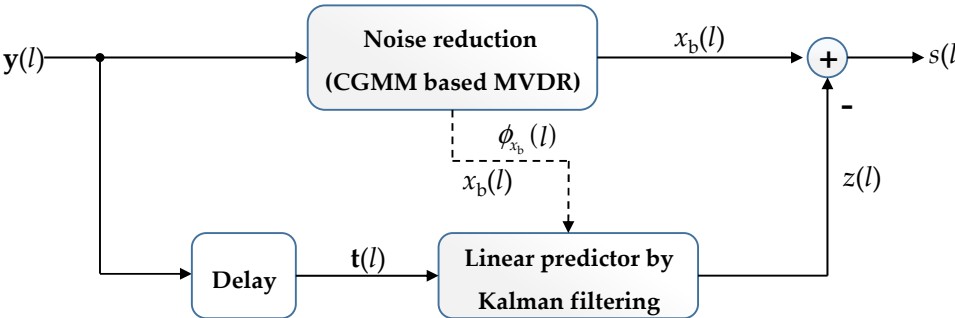

**Figure 1.** Block diagram of the proposed dereverberation method.

### 3.1. Algorithm Architecture

In the upper path of Figure 1, by using Equations (2), (3) and (15), the output $x_b$ (*l*) of beamforming is obtained as follows:

$$x_b(l) = \mathbf{w}_b^H \mathbf{y}(l) = x_e(l) + \mathbf{w}_b^H[\mathbf{r}(l) + \mathbf{v}(l)] \tag{17}$$

Although most of the noise can be suppressed by the MVDR beamformer, the effect of dereverberation is very limited [27]. The scalar $x_e$ (*l*) denotes the inner product $\mathbf{w}_b{}^H \mathbf{x}_e$ (*l*).

In the lower path of Figure 1, the past $(L - D)$ frames are used as the input signal of the linear predictor. By combining the vectors from $\mathbf{y}$ ($l - D$) to $\mathbf{y}$ ($l - L + 1$), the input matrix of the linear predictor at the $l^{th}$ frame is given as follows:

$$\mathbf{t}(l) = \left[ \mathbf{y}^T(l - D), \cdots, \mathbf{y}^T(l - L + 1) \right]^T \tag{18}$$

and the prediction signal $z(l)$ can be expressed as follows:

$$z(l) = \mathbf{w}_r^H(l)\mathbf{t}(l) \tag{19}$$

where $\mathbf{w}_r(l)$ is the linear prediction filter. An appropriate $D$ should be selected according to the frame shift in the STFT operation in order to make the correlation between the prediction error $x_e(l)$ and the prediction signal $z(l)$ zero, i.e.,

$$\langle z, x_e \rangle = \sum_l z(l)x_e(l) = 0 \tag{20}$$

The choice of $D$ determines the amount of early reverberation preserved in the desired signal, which affects the performance of the algorithm [15].

According to the architecture of the proposed method, the final enhanced speech $s(l)$ is obtained as follows:

$$s(l) = x_b(l) - \mathbf{w}_r^H(l)\mathbf{t}(l) \tag{21}$$

where the output $s$ (*l*) is regarded as an estimation of $x_e$ (*l*). If $\mathbf{w}_r$ (*l*) is available, we can estimate the final enhanced signal $s$ (*l*) from Equation (21). In the ISCLP method [12], the Kalman filter was used to jointly estimate the parameters of the sidelobe canceller (SC) and the linear predictor. Since the linear predictor needs information from the past frames, the canceller and predictor cannot work on the same frame simultaneously. It does not mean they are mutually independent due to the common inclusion of the late reverberation components. This implies that changes in the canceller parameters cause changes in the predictor parameters, and vice versa. To avoid this problem, in this paper, the coefficients of the beamformer and the coefficients of the linear predictor are estimated separately. Thus, the result of the beamforming will improve the result of the linear prediction as well.

### 3.2. Application of Kalman Filter

The linear prediction coefficients can be estimated effectively by the Kalman filter [28–30]. In the ISCLP [12], the Kalman filter used for estimating the linear prediction coefficients has shown good results for dereverberation. Here, we define the observation equation of the Kalman filter [12] as follows:

$$x_{\mathrm{b}}^*(l) = \mathbf{t}^H(l)\mathbf{w}_{\mathrm{r}}(l) + x_{\mathrm{e}}^*(l) \tag{22}$$

where the superscript "*" indicates the conjugation operation. The vector $\mathbf{w}_{\mathrm{r}}(l)$ is the state vector with zero mean, and its related covariance matrix is expressed as follows:

$$\mathbf{\Phi}_{\mathrm{w}}(l) = \mathrm{E}\left[\mathbf{w}_{\mathrm{r}}(l)\mathbf{w}_{\mathrm{r}}^{H}(l)\right] \tag{23}$$

and $x_{\mathrm{e}}^*(l)$ in Equation (22) is the desired signal; it is also known as the measurement error. Similarly, the measurement error can also be regarded as a Gaussian process with zero mean, as defined by Equation (6), and its power spectral density is $\phi_{x_{\mathrm{e}}}(l) = \mathrm{E}[x_{\mathrm{e}}(l)x_{\mathrm{e}}^*(l)]$.

The change in $\mathbf{w}_{\mathrm{r}}(l)$ means the change in the RIR or the change in the microphone positions, so $\mathbf{w}_{\mathrm{r}}(l)$ needs to be adjusted by the state equation. Considering the first-order Markov process, the state equation can be defined as follows [14]:

$$\mathbf{w}_{\mathrm{r}}(l) = \mathbf{A}^H(l)\mathbf{w}_{\mathrm{r}}(l-1) + \mathbf{v}_{\mathrm{w}}(l) \tag{24}$$

where $\mathbf{A}(l)$ is the state transition matrix; it indicates the prediction result of the state vectors from the previous frames to the current frame. The zero-mean Gaussian model with the following covariance matrix can be used to model the process noise $\mathbf{v}_{\mathrm{w}}(l)$:

$$\mathbf{\Phi}_{\mathrm{v}}(l) = \mathrm{E}\left[\mathbf{v}_{\mathrm{w}}(l)\mathbf{v}_{\mathrm{w}}^{H}(l)\right] \tag{25}$$

According to Equations (22) and (24), the purpose of the Kalman filter is to find the optimal solution for the linear prediction coefficients based on the minimum mean square error (MMSE). Usually, $\mathbf{A}(l)$ and $\mathbf{v}_{\mathrm{w}}(l)$ will be adjusted at the beginning of the algorithm [31].

In the Kalman filtering process, the essence of the algorithm is to determine the required parameters step by step through two stages. In the first stage, the state vector $\mathbf{w}_{\mathrm{r}}(l)$ of the next moment is predicted through the state equation. This prediction is called a priori time update. In the second stage, the prediction result of the state vector in the first stage is weighted with the so-called Kalman gain that is calculated by the observation equation combined with the state vector predicted in the first stage. The expected optimal state vector $\mathbf{w}_{\mathrm{r}}(l)$ can be obtained by the iterations of these two stages.

Let $\hat{\mathbf{w}}_{\mathrm{r}}(l)$ be the estimated result given by the state Equation (24) and $\hat{\mathbf{w}}_{\mathrm{r}}^+(l)$ be the modified result from the observation Equation (22); their corresponding errors are obtained as follows:

$$\mathbf{e}(l) = \mathbf{w}_{\mathrm{r}}(l) - \hat{\mathbf{w}}_{\mathrm{r}}(l) \tag{26}$$

$$\mathbf{e}^+(l) = \mathbf{w}_{\mathrm{r}}(l) - \hat{\mathbf{w}}_{\mathrm{r}}^+(l) \tag{27}$$

the covariance matrices of $\mathbf{e}(l)$ and $\mathbf{e}^+(l)$ are given as follows:

$$\mathbf{\Phi}_e(l) = \mathrm{E}\left[\mathbf{e}(l)\mathbf{e}^{\mathrm{H}}(l)\right] \tag{28}$$

$$\mathbf{\Phi}_e^+(l) = \mathrm{E}\left[\mathbf{e}^+(l)\mathbf{e}^{+\mathrm{H}}(l)\right] \tag{29}$$

It is worth noting that these two covariance matrices of the errors can be regarded as the evolution in the same frame, that is, $\mathbf{\Phi}_e^+(l)$ is a further estimation of $\mathbf{\Phi}_e(l)$. According to the following cost function:

$$\mathbf{J}\left\{\mathbf{w}_r(l), \hat{\mathbf{w}}_r^+(l)\right\} = \text{tr}\left\{\text{E}\left[\mathbf{e}^+(l)\mathbf{e}^+(l)^H\right]\right\} \tag{30}$$

the update process and the estimation of gain $\mathbf{k}(l)$ of the Kalman filter are expressed as follows [12]:

$$\hat{\mathbf{w}}_r(l) = \mathbf{A}(l)\hat{\mathbf{w}}_r^+(l-1) \tag{31}$$

$$\mathbf{\Phi}_e(l) = \mathbf{A}^H(l)\mathbf{\Phi}_e^+(l-1)\mathbf{A}(l) + \mathbf{\Phi}_v(l) \tag{32}$$

$$s^*(l) = x_b^*(l) - \mathbf{t}^H(l)\hat{\mathbf{w}}_r(l) \tag{33}$$

$$\phi_s(l) = \mathbf{t}^H(l)\mathbf{\Phi}_e(l)\mathbf{t}(l) + \phi_{x_e}(l) \tag{34}$$

$$\mathbf{k}(l) = \mathbf{\Phi}_e(l)\mathbf{t}(l)\phi_s^{-1}(l) \tag{35}$$

$$\hat{\mathbf{w}}_r^+(l) = \hat{\mathbf{w}}_r(l) + \mathbf{k}(l)s^*(l) \tag{36}$$

$$\mathbf{\Phi}_e^+(l) = \mathbf{\Phi}_e(l) - \mathbf{k}(l)\mathbf{t}^H(l)\mathbf{\Phi}_e(l) \tag{37}$$

where Equations (31) and (32) are used to update the predictive coefficients and the co-variance matrix of the errors in the current frame based on the information from the previous frame. In the following, the complex conjugate error signal $s^*(l)$ (i.e., the final enhanced signal) and its power spectral density $\phi_s(l)$ are calculated using Equations (33) and (34) by combining the observation Equation (22) for calculating the Kalman gain $\mathbf{k}(l)$ in Equation (35). The final predictive coefficients and covariance matrix of the errors in the current frame are finally obtained by Equations (36) and (37) with the Kalman gain $\mathbf{k}(l)$. The final results are also used to calculate the information needed for the next frame. In the calculation of the next frame, however, the error signal $s(l)$ and the Kalman gain are reset, which means that they only represent the current frame and do not iterate through the frames. The complex conjugate error signal $s^*(l)$ is the signal we need, which is given by the Kalman filter. The initialization of the Kalman filter is introduced in the following content.

### 3.3. Low-Complexity Algorithm Based on the Kronecker Product

The Kalman filter has the disadvantage of high computational complexity. By observing update Equations (31)~(37) of the Kalman filter, it can be found that the calculation of the Kalman filter involves the multiplication and summation of many matrices with high dimension data, so the dimension reduction in the matrix is the key for reducing the calculation complexity. In the observation Equation (22), the stacked microphone signal $\mathbf{t}(l)$ is a vector with $M(L-D)$ elements, $\mathbf{w}_r(l)$ is a vector including $M(L-D)$ linear prediction coefficients. According to the method proposed in [32], $\mathbf{w}_r(l)$ can be expressed as the form of the Kronecker product of two subfilters, where two low-dimension subfilters can be estimated separately. Considering a signal model such as Equation (21), we have

$$x_e(l) = \mathbf{w}_2^H(l)\mathbf{y}(l) - [\mathbf{w}_1(l) \otimes \mathbf{w}_2(l)]^H\mathbf{t}(l) \tag{38}$$

where $x_e(l)$ is the desired signal, symbol "$\otimes$" represents the Kronecker product, $\hat{x}(l) = \mathbf{w}_2^H(l)\mathbf{y}(l)$, $\mathbf{w}_r(l) = \mathbf{w}_1(l) \otimes \mathbf{w}_2(l)$, $\mathbf{w}_1(l)$, and $\mathbf{w}_2(l)$ can be regarded as a $(L-D)$-order interframe filter and a $M$-order spatial filter, respectively. In fact, $\mathbf{w}_1(l)$ represents the filter for processing the consecutive frames, and the main purpose of $\mathbf{w}_1(l)$ is to predict the reverberation components from the past $L-D$ frames. $\mathbf{w}_2(l)$ represents the filter that processes signal from all channels. Thus, $\mathbf{w}_2(l)$ can be regarded as the coefficients of the spatial filter. By using the Kronecker product, the original optimization problem becomes two optimal problems of the subfilters.

According to the characteristic of the Kronecker product [32], we can obtain

$$\mathbf{w}_1(l) \otimes \mathbf{w}_2(l) = [\mathbf{I} \otimes \mathbf{w}_2(l)]\mathbf{w}_1(l) = \mathbf{W}_2(l)\mathbf{w}_1(l) \tag{39}$$

where $\mathbf{I}$ is a $(L - D) \times (L - D)$ identity matrix and $\mathbf{W}_2(l) = \mathbf{I} \otimes \mathbf{w}_2(l)$ is a $M(L - D) \times (L - D)$ matrix. Substituting (39) into (38), we have

$$x_e(l) = \mathbf{w}_2^H(l)\mathbf{y}(l) - \mathbf{w}_1^H(l)\mathbf{W}_2^H(l)\mathbf{t}(l) = \hat{x}(l) - \mathbf{w}_1^H(l)\mathbf{t}_{W_2}(l) \tag{40}$$

where $\mathbf{t}_{W_2}(l) = \mathbf{W}_2^H(l)\mathbf{t}(l)$.

Since $\mathbf{w}_2(l)$ is a spatial filter, $\mathbf{w}_2(l)$ in (40) can be replaced by the CGMM-MVDR filter, i.e.,

$$x_e(l) = \mathbf{w}_b^H\mathbf{y}(l) - \mathbf{w}_1^H(l)\mathbf{W}^H\mathbf{t}(l) = x_b(l) - \mathbf{w}_1^H(l)\mathbf{t}_W(l) \tag{41}$$

where $\mathbf{w}_b = \mathbf{w}_2(l)$, $\mathbf{W} = (\mathbf{I} \otimes \mathbf{w}_b)$, $\mathbf{t}_w(l) = \mathbf{W}^H(l)\mathbf{t}(l)$. It is worth noting that $\mathbf{w}_b$ is different at each frequency bin of all the frames. $\mathbf{t}_w(l)$ can be regarded as the speech signal that was processed by the spatial filter in advance. After such replacement, two optimal problems of the subfilters become a single optimal problem of the interframe filter that can be estimated by the Kalman filter. Moreover, the state equation and the observation equation of the improved low-complexity Kalman filter are given as follows:

$$\mathbf{w}_1(l) = \mathbf{A}^H(l)\mathbf{w}_1(l - 1) + \mathbf{w}_\Delta(l) \tag{42}$$

and

$$x_b^*(l) = \mathbf{t}_w^H(l)\mathbf{w}_1(l) + x_e^*(l) \tag{43}$$

where the process noise $\mathbf{w}_\Delta(l)$ can be modeled by the zero-mean Gaussian process with the covariance matrix $\mathbf{\Phi}_{w_\Delta}(l) = \mathrm{E}[\mathbf{w}_\Delta(l)\mathbf{w}_\Delta^H(l)]$.

Let $\hat{\mathbf{w}}_1(l)$ be the estimated result by the state Equation (42) and $\hat{\mathbf{w}}_1^+(l)$ be the modified result with the observation Equation (43); their corresponding errors are obtained as follows:

$$\mathbf{e}_1(l) = \mathbf{w}_1(l) - \hat{\mathbf{w}}_1(l) \tag{44}$$

$$\mathbf{e}_1^+(l) = \mathbf{w}_1(l) - \hat{\mathbf{w}}_1^+(l) \tag{45}$$

the covariance matrices of $\mathbf{e}_1(l)$ and $\mathbf{e}_1^+(l)$ are given as follows:

$$\mathbf{\Phi}_{e_1}(l) = \mathrm{E}\left[\mathbf{e}_1(l)\mathbf{e}_1^H(l)\right] \tag{46}$$

$$\mathbf{\Phi}_{e_1}^+(l) = \mathrm{E}\left[\mathbf{e}_1^+(l)\mathbf{e}_1^+(l)^H\right] \tag{47}$$

According to the state Equation (42), the observation Equation (43), and following cost function:

$$\mathbf{J}\{\mathbf{w}_1(l), \hat{\mathbf{w}}_1^+(l)\} = \mathrm{tr}\left\{\mathrm{E}\left[\mathbf{e}_1^+(l)\mathbf{e}_1^+(l)^H\right]\right\} \tag{48}$$

we can obtain update equations for Kalman filtering with low complexity similar to Equations (31)~(37) as follows:

$$\hat{\mathbf{w}}_1(l) = \mathbf{A}(l)\hat{\mathbf{w}}_1^+(l - 1) \tag{49}$$

$$\mathbf{\Phi}_{e_1}(l) = \mathbf{A}^H(l)\mathbf{\Phi}_{e_1}^+(l - 1)\mathbf{A}(l) + \mathbf{\Phi}_{w_\Delta}(l) \tag{50}$$

$$s_1^*(l) = x_b^*(l) - \mathbf{t}_W^H(l)\hat{\mathbf{w}}_1(l) \tag{51}$$

$$\phi_{s_1}(l) = \mathbf{t}_W^H(l)\mathbf{\Phi}_{e_1}(l)\mathbf{t}_W(l) + \phi_{x_e}(l) \tag{52}$$

$$\mathbf{k}_1(l) = \mathbf{\Phi}_{e_1}(l)\mathbf{t}_W(l)\phi_{s_1}^{-1}(l) \tag{53}$$

$$\hat{\mathbf{w}}_1^+(l) = \hat{\mathbf{w}}_1(l) + \mathbf{k}_1(l)s_1^*(l) \tag{54}$$

$$\mathbf{\Phi}_{e_1}^{+}(l) = \mathbf{\Phi}_{e_1}(l) - \mathbf{k}_1(l)\mathbf{t}_{\mathbf{W}}^{H}(l)\mathbf{\Phi}_{e_1}(l) \tag{55}$$

where $s_1^{*}(l)$ is the final enhanced signal by the low-complexity algorithm with power spectral density $\phi_{s_1}(l)$ and $\mathbf{k}_1$ is the Kalman filter gain of the low-complexity algorithm. Thus, the coefficients of the interframe filter $\mathbf{w}_1(l)$ can be estimated. Through Equations from (38) to (55), the dimension of the filter coefficients $\mathbf{w}_r(l)$ is significantly reduced so that the computational complexity is reduced effectively. The effectiveness of the low-complexity algorithm is discussed in the experiments.

### 3.4. Initialization of Kalman Filtering

Some parameters need to be initialized at the start of the Kalman filtering. In update Equations (31)~(37) and (49)~(55), these parameters include the state transition matrix $\mathbf{A}(l)$, covariance matrix $\mathbf{\Phi}_{\mathbf{v}}(l)$ of the process noise $\mathbf{v}_{\mathbf{w}}(l)$, covariance matrix $\mathbf{\Phi}_{\mathbf{w}_{\Delta}}(l)$ of the process noise $\mathbf{w}_{\Delta}(l)$, power spectral density $\phi_{x_e}(l)$ of the desired signal, and the covariance matrices $\mathbf{\Phi}_{e}(l)$ and $\mathbf{\Phi}_{e_1}(l)$ of the errors $\mathbf{e}(l)$ and $\mathbf{e}_1(l)$.

The state transition matrix $\mathbf{A}(l)$ is regarded as a time-invariant function similar to the forgetting factor in the update equation [31]. The covariance matrices $\mathbf{\Phi}_{e}(l)$ and $\mathbf{\Phi}_{e_1}(l)$ gradually become convergent during the Kalman filtering. The covariance matrix $\mathbf{\Phi}_{e}(l)$ can be initialized as a diagonal matrix, i.e., Diag $[\mathbf{\Phi}_{e}(l)] = \mathbf{\varphi}_{e}$, where vector $\mathbf{\varphi}_{e}$ includes $M$ $(L-D)$ elements that rise exponentially with every $M$ elements [12], i.e.,

$$\mathbf{\varphi}_{e} = \left[\varphi_{e}^{1}\mathbf{1}^{T}, \varphi_{e}^{2}\mathbf{1}^{T}, \cdots, \varphi_{e}^{(L-D)}\mathbf{1}^{T}\right]_{M(L-D)\times 1}^{T} \tag{56}$$

where $\mathbf{1} = [1, 1, \cdots, 1]_{M\times 1}^{T}$, $\varphi_{e}^{1}$, $\varphi_{e}^{2}$, $\ldots$, $\varphi_{e}^{(L-D)}$ are $L-D$ scalars that rise exponentially. Similarly, the covariance matrix $\mathbf{\Phi}_{e1}(l)$ is also initialized as a diagonal matrix, i.e., Diag $[\mathbf{\Phi}_{e1}(l)] = \mathbf{\varphi}_{e1}$, where vector $\mathbf{\varphi}_{e1}$ only includes $L-D$ elements that drop exponentially, i.e.,

$$\mathbf{\varphi}_{e1} = \left[\varphi_{e1}^{1}, \varphi_{e1}^{2}, \cdots, \varphi_{e1}^{(L-D)}\right]^{T} \tag{57}$$

The process noises $\mathbf{v}_{\mathbf{w}}(l)$ and $\mathbf{w}_{\Delta}(l)$ are modeled by the zero-mean Gaussian model, and their covariance matrixes $\mathbf{\Phi}_{\mathbf{v}}(l)$ and $\mathbf{\Phi}_{\mathbf{w}\Delta}(l)$ are usually assumed to be time-invariant and initialized as follows [12]:

$$\mathbf{\Phi}_{\mathbf{v}}(l) = (1 - \beta)\mathbf{\Phi}_{e}(l) \tag{58}$$

$$\mathbf{\Phi}_{\mathbf{w}\Delta}(l) = (1 - \beta)\mathbf{\Phi}_{e_1}(l) \tag{59}$$

where $\beta \in (0,1)$ is the forgetting factor.

The weighted prediction error (WPE) algorithm also needs to initialize the PSD of the desired signal. In the WPE, the PSD of unprocessed signal is regarded as one of the desired signals to estimate the linear prediction coefficients used for obtaining the enhanced signal. Then, the PSD of the enhanced signal is used to estimate the new linear prediction coefficients. The above steps are iterated until the algorithm converges. When the WPE is used for a short-time speech signal, the frame number of the speech signal has to be reduced. In this case, the estimated PSD of the desired signal greatly degrades the performance of the algorithm because of using few frames. Therefore, a method of directly estimating the PSD of the desired signal from the reverberated speech by using the DNN was proposed [33]. In [14,15], the PSD of the desired signal is determined by a decision-oriented method, that is, the current PSD of the desired signal is composed of the PSD of the previous frame and the PSD of the current frame; the later one is obtained by the linear prediction coefficients of the previous frame. These two PSDs are weighted by a weight factor. This method relies on the setting of the weight factor, and how to choose an appropriate weight factor is crucial.

In this paper, the speech signal processed by the beamformer is utilized to initialize the PSD of the desired signal. We can redefine the covariance matrices presented in (11) and (12) at each time–frequency bin as follows:

$$\mathbf{C}^{(x+r+n)}(l) = \mathbf{y}(l)\mathbf{y}(l)^H \tag{60}$$

$$\mathbf{C}^{(n)}(l) = \lambda(l)^{(n)}\mathbf{y}(l)\mathbf{y}(l)^H \tag{61}$$

where $\lambda(l)^{(n)}$ denotes the mask or probability of the scenarios that only contain noise. Thus, the covariance matrix of the noiseless speech at each time–frequency bin can be given as follows:

$$\mathbf{C}^{(x+r)}(l) = \mathbf{C}^{(x+r+n)}(l) - \mathbf{C}^{(n)}(l) \tag{62}$$

Obviously, the diagonal elements of the covariance matrix $\mathbf{C}^{(x+r)}(l)$ represent the PSD of the speech signal. We use vector $\boldsymbol{\theta}(l)$ to express its diagonal elements as follows:

$$\boldsymbol{\theta}(l) = \mathrm{Diag}\left[\mathbf{C}^{(x+r)}(l)\right] = \left[\phi_{x_{b1}}(l), \phi_{x_{b2}}(l), \cdots, \phi_{x_{bm}}(l), \cdots, \phi_{x_{bM}}(l)\right]^T \tag{63}$$

where $\phi_{x_{bm}}(l)$ represents the power spectral density of the $m$th microphone signal that is processed by the beamformer. The diagonal elements are summed and averaged as the approximation of the PSD of the desired signal required in the Kalman filtering process, i.e.,

$$\phi_{x_b}(l) = \frac{1}{M}\sum_{m=1}^{M}\phi_{x_{bm}}(l) \tag{64}$$

where $\phi_{x_b}(l)$ is an approximation of the power spectral density $\phi_{x_e}(l)$ of the desired signal, and the low-complexity version also utilize $\phi_{x_b}(l)$ as the initialization of $\phi_{x_e}(l)$. Under ideal circumstances, the closer the initialized PSD is to the PSD of the desired speech, the better the algorithm will achieve, but this is obviously impractical. Blind estimation of the PSD of the desired signal is very difficult only from the signal received by the microphones. Compared with the PSD estimation method in ISCLP [12], the proposed method does not need the incident angle of the desired speech and other prior information. Applying such crude approximation does not mean a wrong result, because the linear prediction coefficients are used instead of the desired signal itself in the Kalman filtering, the effect of the approximation error can be mitigated during the Kalman filtering. Meanwhile, the computational complexity of this kind of PSD estimation is lower than that used in ISCLP. Therefore, it is feasible to adopt this approximation of the PSD, which is also confirmed by the following experiments.

## 4. Experiments and Evaluation

In this section, we give the experimental results. The proposed dereverberation method is compared with the reference methods including ISCLP and WPE, which have excellent performance at present. Firstly, we introduce the experimental settings used for the evaluation. Then, the reference methods are introduced briefly. Finally, the experimental results are evaluated and discussed.

### 4.1. Acoustic Scenario and Experimental Setup

The observation signals are generated the same as described in [34], and the TIMIT corpus [35] is used for the simulation. As shown in Figure 2, the length, width, and height of the room are 4, 3, and 3 m, respectively. A uniform linear microphone array with eight microphones is placed on a table for receiving speech signals from all angles in the room, the microphone spacing is 0.04 m, and the linear microphone array has a height of 0.75 m with the table. Meanwhile, there are 20 speech source positions in the room. Eight source positions, namely Tar.1, Tar.2, Tar.3, Tar.6, Tar.7, Tar.8, Tar.10, and Tar. 20, are chosen for the test (Figure 2), and 100 utterances are selected for each position. The selection of the source positions is based on their symmetry in Figure 2, that is, Tar.1 and Tar.2 are similar

to Tar.5 and Tar.4, Tar.15 and Tar.16, Tar.19 and Tar.18, respectively. Tar.3 is similar to Tar.17. Tar.6 is similar to Tar.11. Tar.7 is similar to Tar.9, Tar.12 and Tar.14. Tar.8 is similar to Tar.13. These eight source positions are represented in Figure 2. Babble noise is used to obtain the reverberate speech with noise, the input signal-to-noise ratio (SNR) is set to −5, 0, 5, and 10 dB, and the reverberation time $T_{60}$ is set to 100, 400, 500, 600, 700, and 800 ms. When investigating the dependence of the algorithm on the order $L$ of the linear prediction filter and the effect of the different initializations of the PSD on the proposed method, we selected the speech signal from the first position as the evaluation object. When comparing with the reference methods, all 800 utterances of the eight aforementioned locations in Figure 2 were utilized.

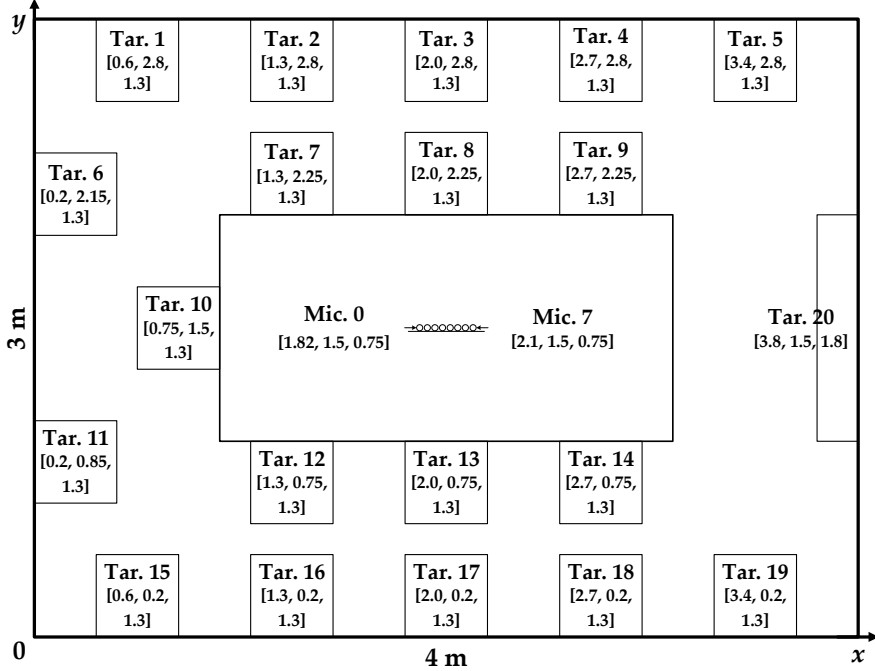

**Figure 2.** Simulation scene of a room considering 20 source locations and 8 microphones [34].

The speech signal is sampled at 16 kHz, a Hanning window with a length of 512 samples has 50% overlap for analyzing the speech signal, and a 512-point STFT is used for the windowed speech signal. The frame number for each batch in the MVDR beamforming is set to 70. The delay $D$ that preserves the early reflections is set to 1, and the order $L$ of the linear prediction filter is set to 10. The state transition matrix $\mathbf{A}$ $(l) = \alpha\mathbf{I}$, where $\alpha = 0.998$. Similar to the default setting in [12], we let $10 \log_{10}(\varphi_e) = -4$ dB and $10 \log_{10}(\varphi_{e1}) = -4$ dB in (56) and (57), whereas $10 \log_{10}(1 - \beta) = -25$ dB in (58) and (59).

To evaluate the proposed algorithm, we choose the perceptual evaluation of speech quality (PESQ) [36], the short-time objective intelligibility (STOI) [37], the cepstral distance (CD) [38], the frequency-weighted segmental SNR (fwSegSNR) [39], and the signal-to-interference ratio (SIR) [39] as the measures, where the reverberation and noise are considered as the interference. These measures have a reasonable evaluation of the perception of reverberation and the overall quality of speech [40]. The CD focuses on reflecting the distortion of the speech spectrum: the lower the CD, the less overall speech distortion. The PESQ, STOI, fwSegSNR, and SIR can better demonstrate the reduction in reverberation and noise.

### 4.2. Reference Methods

To demonstrate the effectiveness of the proposed method, the ISCLP and WPE methods that have excellent performance at present are chosen for the comparison.

ISCLP realized in [41] is a type of method for noise reduction and dereverberation by combining the MCLP and GSC. In the ISCLP, the filter coefficients of the MCLP and GSC are simultaneously estimated by the Kalman filter. In the case of an interfering speaker, the ISCLP can effectively eliminate the interfering speaker, but at the same time it also introduces distortion to the target speaker.

WPE realized in [42] is a type of dereverberation method based on linear prediction and is the most widely used dereverberation algorithm as well. It assumes that the speech signal follows the complex Gaussian distribution with time variance. Its model parameters and linear prediction filter coefficients can be obtained iteratively through maximum likelihood estimation. This algorithm has good suppression effect on reverberation, but it is easily interfered by the noise.

### 4.3. Analysis and Comparison of the Test Results

#### 4.3.1. Effect of Filter Order

The effect of the filter order $L$ on the algorithm performance is investigated in an exemplary simulation condition, namely, SNR = 10 dB and $T_{60}$ = 800 ms. Here, the proposed method and ISCLP are evaluated based on the variation in $L$. Figure 3 shows the incremental results of the PESQ, STOI, CD, and SIR for different filter orders. The ISCLP shows better performance with respect to the PESQ, STOI, and CD in the case of shorter filter order, while for SIR, the ISCLP shows the dependence on the filter order. The proposed method shows the dependence on the filter order in the PESQ, STOI, and SIR results, and its performance is the best at eight options of the filter order and tends to be convergent at some fixed order $L$ for different tests. In the evaluation results of the PESQ, STOI, and CD, the proposed method is better than ISCLP at all filter orders. In the evaluation result of the SIR, the proposed method is slightly worse than ISCLP. It can be seen from Figure 3 that the CD results of the proposed method and ISCLP have positive increments, which indicates that the speech distortion is increased. This may be caused by the inaccurate estimation of some key parameters in the Kalman filtering, but the distortion caused by the ISCLP is significantly larger than the proposed method.

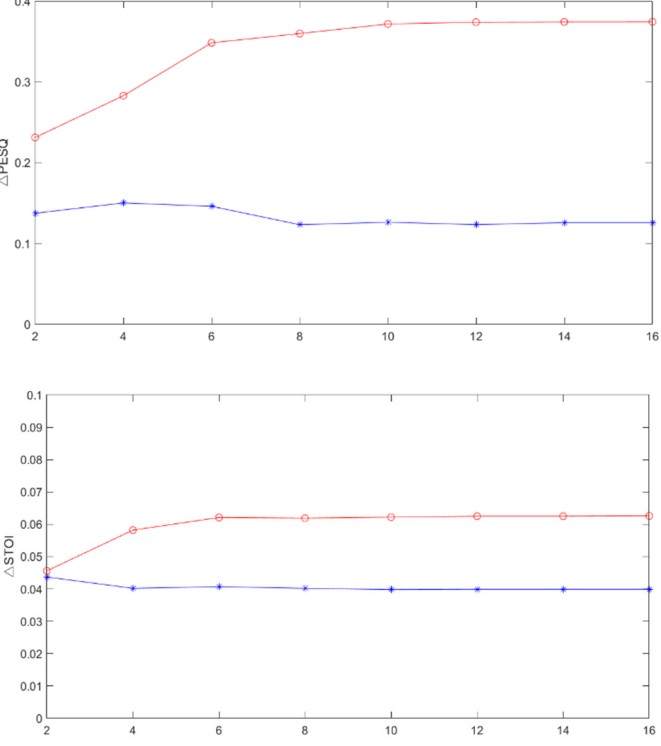

**Figure 3.** *Cont.*

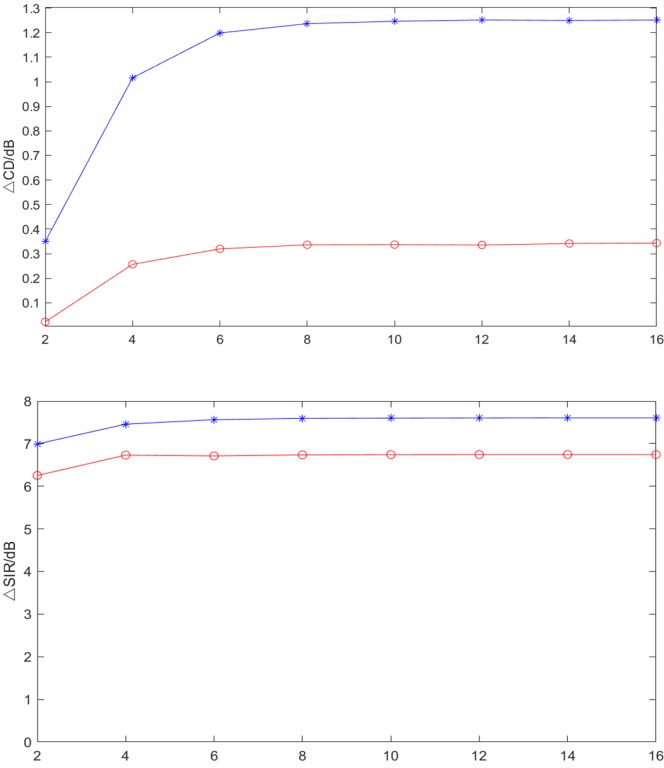

**Figure 3.** Test results related to the filter order *L* between the proposed method (red line) and ISCLP (blue line).

### 4.3.2. Effect of the PSD Initialization

We measured the effect of different initialization methods of the PSD on the proposed method when the input $SNR = 10$ dB and $T_{60} = \{400, 500, 600, 700, 800\}$ ms at the same source positions as given in Section 4.3.1. Four initialization methods of the PSD of the desired signal, i.e., the method given in ISCLP, the proposed initialization method, the method given in [33], and the method given in [14,15], are employed in the proposed dereverberation algorithm. The effects of these methods on the performance of the algorithms are compared. In addition, the initialization of the PSD using real PSD of the desired signal is also compared. Figure 4 shows the result comparison.

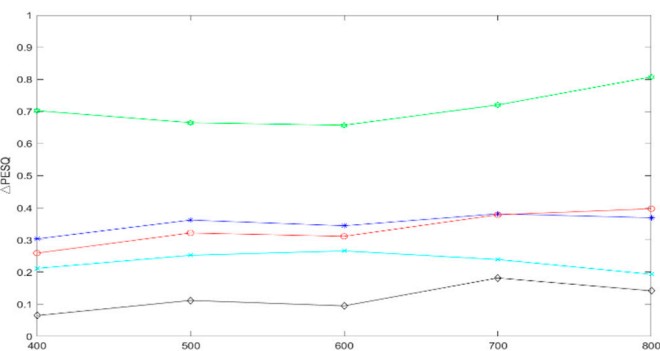

**Figure 4.** *Cont.*

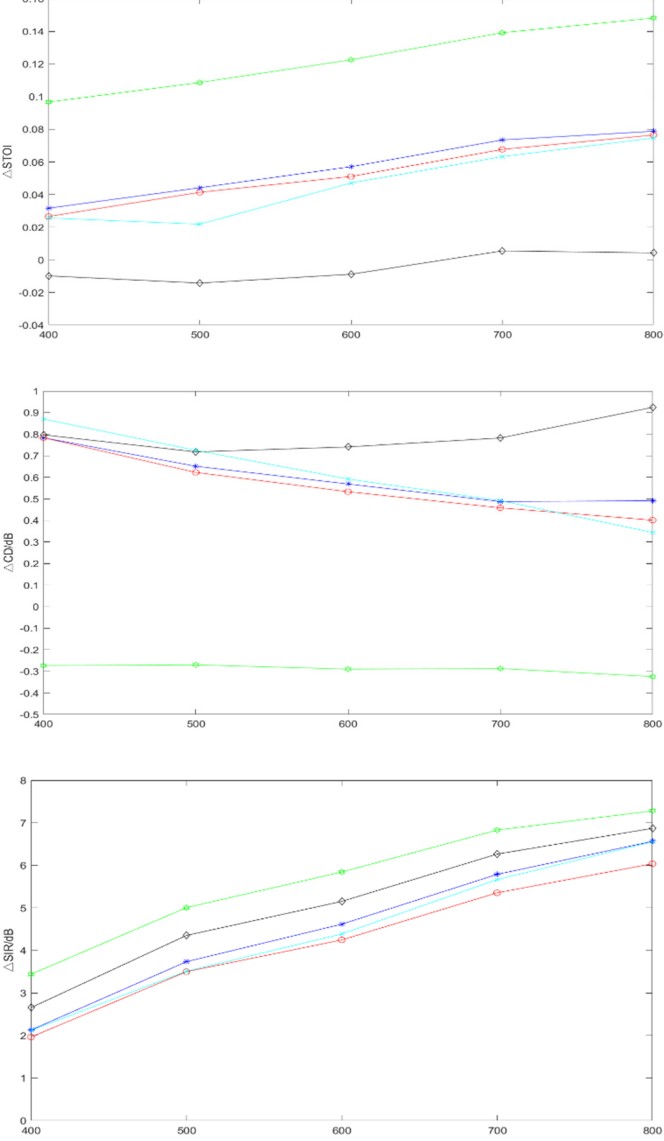

**Figure 4.** Test results related to $T_{60}$ on four initialization methods of the PSD of the desired signal with real PSD (green line), [14,15] (black line), proposed (red line), ISCLP (blue line), and [34] (cyan line).

As expected, the result of using real PSD of the desired signal greatly improved the performance, which is the best among four evaluation indicators. The effect of the initialization of the ISCLP and the proposed method are far better than the initialization given in [33] and [14,15]. In terms of SIR, the effects of these four initialization methods (excluding the real PSD) are very similar.

The effect of the initialization of the ISCLP is slightly better than the proposed method in the results of the PESQ, STOI, and SIR, but the ISCLP needs prior information such as the incident angle of the desired signal and the position of the microphones in advance. In the ISCLP, the inaccurate incident angle of the desired signal makes the block matrix in the GSC filter miss useful signal, which causes the distortion of the desired signal in the subsequent processing. The proposed method only uses the signals received by the microphones to obtain the estimation of the PSD of the desired signal, which is an adaptive approach. At the same time, the computational complexity of the initialization of the PSD in the proposed method is far lower than that of the ISCLP. From the experimental results, we can find that obtaining the PSD close to the real PSD of the desired signal is an

important way for effectively improving the performance of the algorithm. How to obtain the accurate PSD of the desired signal is a key issue for the dereverberation.

### 4.3.3. Performance Comparison with Reference Methods

Two proposed dereverberation algorithms and two reference methods are evaluated for different *SNR* and $T_{60}$. Six kinds of reverberation time are considered at four *SNR* conditions. Tables 1–4 show the results of performance evaluation for the unprocessed speech. It is worth noting that the PESQ value of the unprocessed signal increases with the increase in the reverberation time. This is because in the case of lower input *SNR*, compared with low reverberation, the high reverberation can resist more noise, so that the speech is not submerged into the annoying noise. Under the condition of high SNR, the performance of the proposed method is not very good compared with the reference methods, and especially worse than the WPE. The reason is that under the condition of high SNR, the speech distortion caused by the noise is far less than that caused by the reverberation. Meanwhile, WPE can remove reverberation well and improve the speech quality greatly. Under the condition of low reverberation, the ISCLP shows better results and the WPE shows worse results, because WPE has no denoising ability. Under the condition of low SNR, it can be found that the results of the PESQ, STOI, and fwSegSNR of the proposed method are higher than these two reference methods in most cases, and the result of STOI is significantly improved especially in the case of $-5$ dB, whereas the low-complexity version of the proposed method (called proposed-kron) is similar to the proposed method. As shown in Tables 1 and 2, the fwSegSNR is also significantly improved in all cases. In the results for CD, the WPE shows the best performance. As the distortion measurement of the speech, the results of CD show that both the proposed method and ISCLP introduce the distortion, and the ISCLP causes more distortion. The distortion is due to the PSD error of the desired signal in the Kalman filtering. The GSC-based beamforming used in the ISCLP causes speech distortion because of the inaccurate RETF. For the SIR, the ISCLP shows the best effect. The ISCLP reduces much of the interference at the cost of introducing more speech distortion. The proposed method balances the speech distortion and interference reduction. Within a certain distortion range, the proposed method eliminates more interference signal than the WPE. The proposed-kron method also shows good results with low complexity. From the discussion above, the proposed method shows better performance under the condition of low SNR and high reverberation. In future work, the problem of speech distortion in the proposed method needs to be improved.

**Table 1.** Comparison of the test results at $-5$ dB for babble noise. (The bold in the table indicates the best result).

| $T_{60}$ (ms) | 100 | 400 | 500 | 600 | 700 | 800 |
|---|---|---|---|---|---|---|
| | | | | PESQ | | |
| unprocessed | 1.12 | 1.51 | 1.56 | 1.58 | 1.61 | 1.62 |
| ISCLP | **1.50** | 1.58 | 1.63 | 1.66 | 1.69 | 1.71 |
| WPE | 1.12 | 1.47 | 1.52 | 1.53 | 1.54 | 1.56 |
| proposed | 1.20 | 1.66 | 1.72 | **1.77** | **1.78** | **1.79** |
| proposed-kron | 1.19 | **1.68** | **1.75** | **1.77** | **1.78** | **1.79** |
| | | | | STOI | | |
| unprocessed | 0.51 | 0.58 | 0.58 | 0.58 | 0.56 | 0.56 |
| ISCLP | **0.59** | 0.57 | 0.57 | 0.57 | 0.57 | 0.57 |
| WPE | 0.50 | 0.57 | 0.57 | 0.58 | 0.59 | 0.59 |
| proposed | 0.54 | **0.61** | **0.62** | **0.62** | **0.62** | **0.62** |
| proposed-kron | 0.54 | **0.61** | **0.62** | **0.62** | **0.62** | 0.61 |

**Table 1.** *Cont.*

| $T_{60}$ (ms) | 100 | 400 | 500 | 600 | 700 | 800 |
|---|---|---|---|---|---|---|
| | | | CD (dB) | | | |
| unprocessed | 7.04 | **5.98** | 5.92 | 5.89 | 5.89 | 5.90 |
| ISCLP | 7.93 | 7.91 | 7.87 | 7.87 | 7.87 | 7.88 |
| WPE | **7.00** | **5.98** | **5.90** | **5.86** | **5.82** | **5.80** |
| proposed | 7.60 | 7.13 | 7.06 | 7.01 | 6.96 | 6.93 |
| proposed-kron | 7.05 | 6.32 | 6.23 | 6.17 | 6.13 | 6.41 |
| | | | SIR (dB) | | | |
| unprocessed | **11.70** | −4.37 | −5.52 | −6.11 | −6.89 | −7.03 |
| ISCLP | 7.02 | **−1.17** | **−1.62** | **−1.13** | **−1.31** | **−0.98** |
| WPE | 6.34 | −1.42 | −2.28 | −2.97 | −3.43 | −3.98 |
| proposed | −2.00 | −2.17 | −2.73 | −2.74 | −2.31 | −1.90 |
| proposed-kron | −3.14 | −3.12 | −3.33 | −3.42 | −3.11 | −2.70 |
| | | | fwSegSNR (dB) | | | |
| unprocessed | 0.65 | 1.93 | 2.11 | 2.22 | 2.30 | 2.35 |
| ISCLP | 1.41 | 1.67 | 1.63 | 1.57 | 1.51 | 1.46 |
| WPE | 0.83 | 2.02 | 2.17 | 2.28 | 2.35 | 2.41 |
| proposed | 3.16 | 4.03 | 4.16 | 4.24 | 4.30 | 4.34 |
| proposed-kron | **3.43** | **4.62** | **4.79** | **4.89** | **4.96** | **5.01** |

**Table 2.** Comparison of the test results at 0 dB for babble noise.

| $T_{60}$ (ms) | 100 | 400 | 500 | 600 | 700 | 800 |
|---|---|---|---|---|---|---|
| | | | PESQ | | | |
| unprocessed | 1.46 | 1.84 | 1.87 | 1.88 | 1.88 | 1.87 |
| ISCLP | **2.02** | 1.84 | 1.88 | 1.90 | 1.90 | 1.90 |
| WPE | 1.49 | 1.79 | 1.83 | 1.86 | 1.88 | 1.91 |
| proposed | 1.51 | 2.01 | 2.06 | **2.10** | **2.11** | **2.11** |
| proposed-kron | 1.45 | **2.06** | **2.09** | **2.10** | 2.09 | 2.08 |
| | | | STOI | | | |
| unprocessed | 0.62 | 0.66 | 0.65 | 0.64 | 0.64 | 0.63 |
| ISCLP | **0.71** | 0.64 | 0.64 | 0.64 | 0.63 | 0.63 |
| WPE | 0.61 | 0.67 | 0.67 | 0.67 | 0.67 | 0.67 |
| proposed | 0.63 | **0.69** | **0.70** | **0.70** | **0.69** | **0.69** |
| proposed-kron | 0.64 | **0.69** | 0.69 | 0.68 | 0.67 | 0.67 |
| | | | CD (dB) | | | |
| unprocessed | 6.42 | 5.27 | 5.27 | 5.30 | 5.35 | 5.40 |
| ISCLP | 7.16 | 7.46 | 7.42 | 7.43 | 7.55 | 7.68 |
| WPE | **6.30** | **5.16** | **5.10** | **5.06** | **5.04** | **5.13** |
| Proposed | 6.97 | 6.28 | 6.32 | 6.34 | 6.35 | 6.37 |
| proposed-kron | 6.40 | 5.38 | 5.47 | 5.56 | 5.66 | 5.75 |
| | | | SIR (dB) | | | |
| unprocessed | **13.41** | −3.80 | −4.80 | −5.59 | −6.11 | −6.51 |
| ISCLP | 8.81 | **−0.11** | **−0.15** | **−0.38** | **−0.41** | **−0.58** |
| WPE | 6.69 | −0.16 | −1.51 | −2.34 | −2.94 | −3.51 |
| proposed | −2.21 | −1.99 | −1.15 | −1.34 | −1.11 | −1.18 |
| proposed-kron | −3.34 | −2.08 | −2.12 | −2.26 | −2.12 | −2.25 |
| | | | fwSegSNR (dB) | | | |
| unprocessed | 1.89 | 3.33 | 3.43 | 3.47 | 3.47 | 3.46 |
| ISCLP | 3.56 | 2.80 | 2.65 | 2.50 | 2.40 | 2.30 |
| WPE | 2.15 | 3.56 | 3.70 | 3.79 | 3.84 | 3.88 |
| proposed | 4.56 | 5.23 | 5.31 | 5.33 | 5.37 | 5.37 |
| proposed-kron | **4.91** | **5.87** | **5.96** | **6.00** | **6.02** | **6.01** |

**Table 3.** Comparison of the test results at 5 dB for babble noise. (The bold in the table indicates the best result).

| $T_{60}$ (ms) | 100 | 400 | 500 | 600 | 700 | 800 |
|---|---|---|---|---|---|---|
| | | | PESQ | | | |
| unprocessed | 1.85 | 2.08 | 2.06 | 2.03 | 2.00 | 1.97 |
| ISCLP | **2.41** | **2.41** | 2.38 | 2.34 | 2.30 | 2.26 |
| WPE | 1.90 | 2.17 | 2.20 | 2.23 | 2.24 | 2.25 |
| proposed | 2.15 | 2.38 | **2.39** | **2.37** | **2.34** | **2.31** |
| proposed-kron | 2.14 | 2.37 | 2.35 | 2.31 | 2.28 | 2.23 |
| | | | STOI | | | |
| unprocessed | 0.73 | 0.72 | 0.71 | 0.69 | 0.68 | 0.66 |
| ISCLP | **0.79** | **0.76** | **0.75** | **0.75** | 0.74 | 0.73 |
| WPE | 0.72 | 0.75 | **0.75** | **0.75** | **0.75** | **0.75** |
| proposed | 0.75 | **0.76** | **0.75** | **0.75** | 0.74 | 0.74 |
| proposed-kron | 0.75 | 0.75 | 0.74 | 0.73 | 0.72 | 0.71 |
| | | | CD (dB) | | | |
| unprocessed | 5.57 | 5.22 | 5.33 | 5.45 | 5.57 | 5.68 |
| ISCLP | 6.42 | 7.07 | 7.15 | 7.21 | 7.28 | 7.34 |
| WPE | **5.39** | **4.87** | **4.82** | **4.80** | **4.79** | **4.78** |
| proposed | 6.39 | 6.74 | 6.68 | 6.64 | 6.60 | 6.59 |
| proposed-kron | 5.59 | 6.01 | 6.00 | 6.00 | 6.00 | 6.02 |
| | | | SIR (dB) | | | |
| unprocessed | **14.13** | −5.48 | −6.53 | −7.26 | −7.80 | −8.21 |
| ISCLP | 9.38 | **−1.02** | **−1.62** | **−2.03** | **−2.29** | **−2.49** |
| WPE | 6.76 | −1.85 | −2.92 | −3.75 | −4.44 | −5.02 |
| proposed | −2.19 | −6.82 | −6.83 | −6.69 | −6.57 | −6.41 |
| proposed-kron | −3.32 | −8.03 | −8.06 | −7.94 | −7.87 | −7.71 |
| | | | fwSegSNR (dB) | | | |
| unprocessed | 3.83 | 4.71 | 4.68 | 4.59 | 4.49 | 4.39 |
| ISCLP | **5.63** | 3.85 | 3.34 | 3.16 | 3.02 | 2.88 |
| WPE | 4.13 | **5.14** | **5.24** | **5.29** | **5.31** | **5.32** |
| proposed | 4.12 | 4.25 | 4.24 | 4.23 | 4.20 | 4.14 |
| proposed-kron | 4.63 | 4.94 | 4.94 | 4.90 | 4.84 | 4.76 |

**Table 4.** Comparison of the test results at 10 dB for babble noise. (The bold in the table indicates the best result).

| $T_{60}$ (ms) | 100 | 400 | 500 | 600 | 700 | 800 |
|---|---|---|---|---|---|---|
| | | | PESQ | | | |
| unprocessed | 2.24 | 2.32 | 2.24 | 2.18 | 2.12 | 1.97 |
| ISCLP | **2.73** | 2.59 | 2.53 | 2.47 | 2.42 | 2.36 |
| WPE | 2.30 | 2.51 | 2.54 | **2.55** | **2.56** | **2.55** |
| proposed | 2.54 | **2.61** | **2.58** | 2.53 | 2.48 | 2.45 |
| proposed-kron | 2.51 | 2.58 | 2.52 | 2.45 | 2.39 | 2.34 |
| | | | STOI | | | |
| unprocessed | 0.82 | 0.76 | 0.74 | 0.72 | 0.70 | 0.69 |
| ISCLP | **0.85** | 0.79 | 0.78 | 0.77 | 0.76 | 0.75 |
| WPE | 0.81 | **0.80** | **0.80** | **0.80** | **0.80** | **0.80** |
| proposed | 0.81 | 0.79 | 0.79 | 0.78 | 0.77 | 0.77 |
| proposed-kron | 0.81 | 0.78 | 0.77 | 0.76 | 0.74 | 0.73 |

**Table 4.** *Cont.*

| $T_{60}$ (ms) | 100 | 400 | 500 | 600 | 700 | 800 |
|---|---|---|---|---|---|---|
| | | | CD (dB) | | | |
| unprocessed | 4.64 | 4.78 | 5.00 | 5.21 | 5.38 | 5.53 |
| ISCLP | 5.80 | 6.73 | 6.84 | 6.94 | 7.02 | 7.10 |
| WPE | **4.41** | **4.20** | **4.19** | **4.20** | **4.21** | **4.21** |
| proposed | 5.64 | 6.29 | 6.27 | 6.26 | 6.25 | 6.24 |
| proposed-kron | 5.32 | 5.71 | 5.75 | 5.79 | 5.82 | 5.85 |
| | | | SIR (dB) | | | |
| unprocessed | **14.19** | −5.53 | −6.67 | −7.32 | −7.81 | −8.62 |
| ISCLP | 9.55 | **−0.98** | **−1.57** | **−1.96** | **−2.21** | **−2.42** |
| WPE | 6.78 | −1.90 | −2.99 | −3.84 | −4.54 | −5.13 |
| proposed | −1.91 | −6.79 | −6.80 | −6.67 | −6.49 | −6.38 |
| proposed-kron | −3.03 | −8.05 | −8.09 | −7.92 | −7.82 | −7.73 |
| | | | fwSegSNR (dB) | | | |
| unprocessed | 6.39 | 5.86 | 5.65 | 5.43 | 5.22 | 5.04 |
| ISCLP | **7.33** | 4.14 | 3.84 | 3.63 | 3.45 | 3.28 |
| WPE | 6.55 | **6.53** | **6.56** | **6.55** | **6.54** | **6.51** |
| proposed | 5.86 | 5.24 | 5.13 | 5.04 | 4.94 | 4.86 |
| proposed-kron | 6.27 | 5.76 | 5.64 | 5.53 | 5.40 | 5.27 |

### 4.3.4. Comparison of the Spectrogram

Figure 5 shows the spectrogram comparison for the original speech, the enhanced speech obtained by the ISCLP with filter order $L = 6$, the enhanced speech obtained by the WPE, and the enhanced speech obtained by the proposed algorithm. The input SNR is set to 10 dB and $T_{60} = 800$ ms. Compared with the original speech, these three algorithms effectively reduce the reverberation and exhibit different characteristics at the same time. As shown in Figure 5b, much of the reverberation and noise are suppressed by the ISCLP, but the distortion of the desired signal is also serious. Figure 5c shows that in the WPE, most of the reverberation is suppressed and the desired speech is retained well, whereas the noise is hardly suppressed. Figure 5d shows that the proposed algorithm can reduce the noise and reverberation effectively, and can retain the high frequency components of the desired speech. Compared with the WPE algorithm, the proposed algorithm has better ability for suppressing the noise.

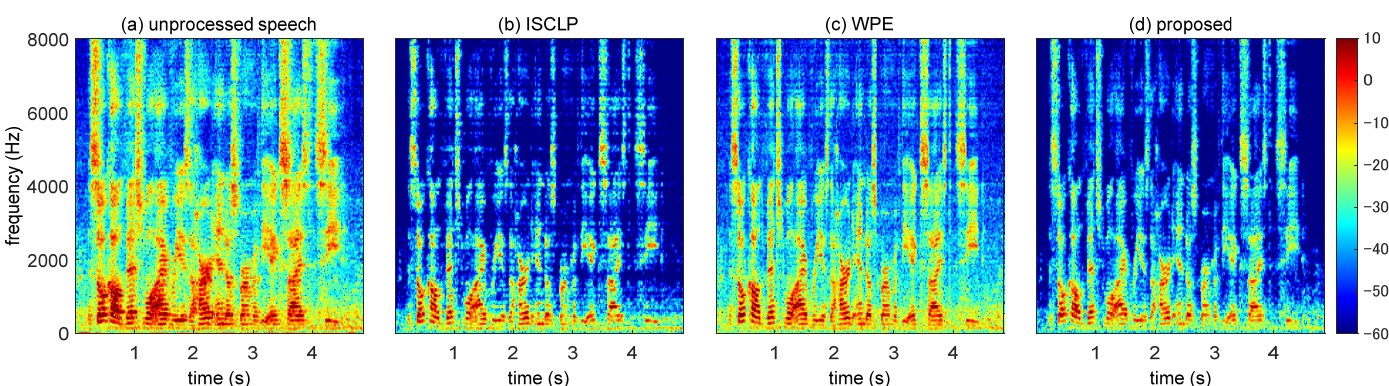

**Figure 5.** Spectrogram comparison in the case of $SNR = 10$ dB and $T_{60} = 800$ ms.

### 4.3.5. Comparison of Computational Complexity

In order to compare the computational complexity of the "proposed" method and the "proposed-kron" method, Table 5 lists the operations required for the multiplication, addition, and division of two proposed algorithms. The update Equations (31), (32), (49),

and (50) of the Kalman filtering are used for the calculation of the first step, whereas update Equations (33)–(35) and (51)–(53) are utilized for the calculation of the second step. Furthermore, the update Equations (36), (37), (54), and (55) are employed for the calculation of the third step. The complexity reduction factor is defined as the ratio between the operations required for the "proposed-kron" and "proposed" methods in each step. We can see that the "proposed-kron" algorithm has greatly improved computational efficiency compared to the "proposed" algorithm. Especially, the multiplication operations in the first and third steps are reduced much more. The "proposed-kron" method not only effectively reduces the computational complexity but also guarantees the speech quality compared to the "proposed" method.

**Table 5.** Complexity comparison.

| Proposed-Kron | | | |
|---|---|---|---|
| Step | $(\times)$ | $(+)$ | $(\div)$ |
| 1 | $(2L - 1)(L - 1)^2$ | $2(L - 1)^3 - (L - 1)$ | $--$ |
| 2 | $M(L - 1)^2(M + 1) + 2(L - 1)^2 + 2(L - 1)$ | $(M^2 + 2)(L - 1)^2 - M(L - 1)$ | $L - 1$ |
| 3 | $(L - 1) + (L - 1)^2 + (L - 1)^3$ | $(L - 1)^3 + (L - 1)$ | $--$ |
| Proposed | | | |
| Step | $(\times)$ | $(+)$ | $(\div)$ |
| 1 | $2M^3(L - 1)^3 + M^2(L - 1)^2$ | $2M^3(L - 1)^3 - M(L - 1)$ | $--$ |
| 2 | $2M^2(L - 1)^2 + 2M(L - 1)$ | $2M^2(L - 1)^2$ | $M(L - 1)$ |
| 3 | $M(L - 1) + M^2(L - 1)^2 + M^3(L - 1)^3$ | $M(L - 1)^3 + (L - 1)$ | $--$ |
| Complexity Reduction Factor | | | |
| Step | $\lambda_{(\times)} \approx$ | $\lambda_{(+)} \approx$ | $\lambda_{(\div)} \approx$ |
| 1 | $1/M^3$ | $1/M^3$ | $--$ |
| 2 | $1/2 + 1/2M$ | $1/2 + 1/M^2$ | $1/M$ |
| 3 | $1/M^3$ | $1/M$ | $--$ |

## 5. Conclusions

In this paper, a noise reduction and dereverberation algorithm was presented by combining the CGMM-based MVDR beamforming and the multichannel linear prediction with Kalman filtering. The noise and a part of reverberation were processed by the MVDR beamforming, meanwhile the covariance matrix of the target speech estimated by beamforming was utilized to estimate the PSD of the target signal, which is an important parameter for Kalman filtering. In addition, a low-complexity version of the proposed algorithm was given. The effectiveness of the proposed method in the experiment, the influence of different PSD initialization methods on the algorithm, and the importance of accurate PSD estimation to the algorithm were carefully analyzed. The evaluation results proved that the proposed algorithm is superior to the reference methods and the low-complexity version keeps similar effectiveness with much less complexity.

**Author Contributions:** Conceptualization, C.B., F.T. and J.Z.; methodology, C.B. and F.T.; software, F.T. and J.Z.; validation, C.B., F.T. and J.Z.; formal analysis, C.B. and F.T.; investigation, F.T.; resources, F.T.; data curation, F.T. and J.Z.; writing—original draft preparation, C.B., F.T. and J.Z.; writing—review and editing, C.B., F.T. and J.Z.; visualization, F.T.; supervision, C.B. All authors have read and agreed to the published version of the manuscript.

**Funding:** This work was funded by the National Natural Science Foundation of China (Grant No. 61831019).

**Institutional Review Board Statement:** Not applicable.

**Informed Consent Statement:** Not applicable.

**Data Availability Statement:** Exclude this statement.

**Acknowledgments:** The authors are grateful to all the reviewers and editors.

**Conflicts of Interest:** The authors declare no conflict of interest.

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
