# Peer review of "Effective Dereverberation with a Lower Complexity at Presence of the Noise"

_applsci, doi:10.3390/app122211819_

Round 1

Reviewer 1 Report

Letter to editor in chief

Dear prof.  Dr. Takayoshi Kobayashi 

I have read the manuscript number: applsci-2044021 with title: "Effective Dereverberation with a Lower Complexity at Presence of the Noise ", and I found that, authors have been tried to reduce the noise and reverberation by combining the MVDR beam former and MCLP. I would like the authors consider the following points before I recommend publication

1- Authors must appear their work, the novelty, and the important results that they obtain only in abstract.

2-In line 1 section 3.1 page 3 the abbreviation of complex Gaussian Mixture Model is "CGMM", so authors must clean the word" model".

3-Authors put a separate section with title "Review of the relevant works ". Why aren't they put it in the introduction?, the manuscript must be rearranged

4-The presentation and grammar need improvement.

Author Response

Dear reviewer,

Thanks a lot for your hard work to review our paper and give us many good comments or suggestions. Based on these important comments or suggestions, we revised the manuscript. In the following, we will response the comments one by one, where the comments are highlighted by the blue color.

 #Comment 1: Authors must appear their work, the novelty, and the important results that they obtain only in abstract.

Answer: Thank you very much for your comment. Actually, we described the novelty of our work and the important results in the abstract, as follows:

Our work: “In this paper, the MVDR beamformer and MCLP are effectively combined for the noise reduction and dereverberation.” (Please refer to lines 12 to 13 of the revised manuscript)

Novelty and contributions: “Especially, the MCLP coefficients are estimated by the Kalman filter, the MVDR filter based on complex Gaussian Mixture Model (CGMM) is used to enhance the speech corrupted by the reverberation with the noise and estimate the power spectral density (PSD) of the target speech required by the Kalman filter, respectively. The finally enhanced speech is obtained by the Kalman filter. Furthermore, a complexity reduction method with respect to the Kalman filter is also proposed based on the Kronecker product.” (Please refer to lines 13 to 19)

Important results: “Compared to two advanced algorithms, the integrated sidelobe cancellation and linear prediction (ISCLP) method, and the weighted prediction error (WPE) method, which are very effective for removing reverberation, the proposed algorithm shows better performance and lower complexity.” (Please refer to lines 19 to 22)

#Comment 2: In line 1 section 3.1 page 3 the abbreviation of complex Gaussian Mixture Model is “CGMM”, so authors must clean the word “model”.

Answer: Thank you very much for your comment. This issue has been revised in the manuscript. (Please refer to line 124 of the revised manuscript)

#Comment 3: Authors put a separate section with title “review of the relevant works”. Why aren’t they put it in the introduction? The manuscript must be rearranged.

Answer: Thank you very much for your advice. The part of “review of the relevant works” includes two signal models, i.e., the CGMM-based MVDR beamforming model and the MCLP reverberation model. Therefore, in the revised manuscript, the structure has been rearranged, i.e., the sections 3.1 and 3.2 are revised as the section 2.1 and section 2.2.

#Comment 4: The presentation and grammar need improvement.

Answer: Thanks for your suggestion. We checked the full manuscript, and modified the relevant presentation and grammar (marked in blue in the revised manuscript).

Reviewer 2 Report

- Equations requires citations and simple explanation.

- Normal distribution is a continuous probability distribution,  and in many places you used the summation.  You have to explain how and why ( eq. 7 and 8 as examples). 

. Equation 5 for example requires re-check and citation. 

- Figure 1 requires analysis about the delay value, filter role, and a relation between the input and the output.

- The title of figure 2 talks about 8 Microphones, while inside the figure you set 10 microphones (mic. 0    to  mic. 9).

- Results can be summarized  with significant comments.

Author Response

Dear reviewer,

Thanks a lot for your hard work to review our paper and give us many good comments or suggestions. Based on these important comments or suggestions, we revised the manuscript. In the following, we will response the comments one by one, where the comments are highlighted by the blue color.

#Comment 1: Equations requires citations and simple explanation.

Answer: Thank you for your comment. We added the citations and explanation to some important equations (marked in blue in the revised manuscript), for example, lines 140 and 151 of the revised manuscript.

#Comment 2: Normal distribution is a continuous probability distribution, and in many places you used the summation. You have to explain how and why (Eq. 7 and Eq. 8 as examples).

Answer: Thank you for your question. As you said, the normal distribution is a continuous probability density function. However, equations 7 and 8 are the sum of the frame index l and the scenarios index d. The normal distribution probability density function will get specific values for l and d, so there is no problem in summing these values.

#Comment 3: Equation 5 for example requires re-check and citation.

Answer: Thank you for your suggestion. We checked the equations of the manuscript, and added the citations to some important equations, such as Eq. (5) . (Please refer to lines 140 to 141 of the revised manuscript)

#Comment 4: Figure 1 requires analysis about the delay value, filter role, and a relation between the input and the output.

Answer: Thank you for your advice. We have analyzed the delay value and filter role of the Figure 1, and explained the relationship between the input and output, please refer to lines 206 to 215 for the specific description in the revised manuscript.

#Comment 5: The title of figure 2 talks about 8 Microphones, while inside the figure you set 10 microphones (Mic. 0 to Mic. 9).

Answer: Thanks for your suggestion. In the revised manuscript, the microphone number of Figure 2 has been modified as 8 microphones.

#Comment 6: Results can be summarized with significant comments.

Answer: Thank you for your advice. In fact, for each experimental result, we have analyzed and described the advantages of our proposed method, such as the lines 551 to 555 of the revised manuscript described the PSD is an important way for effectively improving the performance of the algorithm, and 584 to 586 of the revised manuscript described the advantages of the proposed-kron method.